**Data Availability Statement:** The Rocky Mountain elk genome has been deposited at GenBank accession and associated sequencing reads to the NCBI SRA database under BioProject PRJNA657053. All programs and scripts are

# A pseudomolecule assembly of the Rocky Mountain elk genome

Rick E. Masonbrink[1]*, David Alt[2], Darrell O. Bayles[2], Paola Boggiatto[2], William Edwards[3], Fred Tatum[4], Jeffrey Williams[2], Jennifer Wilson-Welder[2], Aleksey Zimin[5], Andrew Severin[1], Steven Olsen[2]

1 Genome Informatics Facility, Department of Biotech, Iowa State University, Ames, Iowa, United States of America, 2 Infectious Bacterial Diseases Research Unit, National Animal Disease Center, U.S. Department of Agriculture, Agricultural Research Service, Ames, Iowa, United States of America, 3 Wildlife Health Laboratory, Wyoming Game and Fish Department, Laramie, Wyoming, United States of America, 4 Respiratory Diseases Research Unit, National Animal Disease Center, U.S. Department of Agriculture, Agricultural Research Service, Ames, Iowa, United States of America, 5 Department of Biomedical Engineering, Johns Hopkins University, Baltimore, Maryland, United States of America

* remkv6@iastate.edu

## Abstract

Rocky Mountain elk (*Cervus canadensis*) populations have significant economic implications to the cattle industry, as they are a major reservoir for *Brucella abortus* in the Greater Yellowstone area. Vaccination attempts against intracellular bacterial diseases in elk populations have not been successful due to a negligible adaptive cellular immune response. A lack of genomic resources has impeded attempts to better understand why vaccination does not induce protective immunity. To overcome this limitation, PacBio, Illumina, and Hi-C sequencing with a total of 686-fold coverage was used to assemble the elk genome into 35 pseudomolecules. A robust gene annotation was generated resulting in 18,013 gene models and 33,422 mRNAs. The accuracy of the assembly was assessed using synteny to the red deer and cattle genomes identifying several chromosomal rearrangements, fusions and fissions. Because this genome assembly and annotation provide a foundation for genome-enabled exploration of Cervus species, we demonstrate its utility by exploring the conservation of immune system-related genes. We conclude by comparing cattle immune system-related genes to the elk genome, revealing eight putative gene losses in elk.

## Introduction

Rocky Mountain elk (*Cervus canadensis*) were once distributed across much of North America but now inhabit remote areas. Rocky Mountain elk were nearly exterminated from the Rocky Mountains of Alberta and British Columbia in the early 1900s [1], but were restocked between 1916–1920 with elk from the Greater Yellowstone Area [2–5]. By 1940 elk populations expanded so greatly, that periodic culling was necessary [3, 6]. While elk have been reintroduced to many areas, the densest populations are maintained in mountainous remote areas, like the Greater Yellowstone Area.

available at https://github.com/ISUgenomics/elk_genomics.

**Funding:** This work was supported by the USDA National Institute of Food and Agriculture under grant 2018-67015-28199 to AZ.

**Competing interests:** The authors have declared that no competing interests exist.

Elk typically avoid the presence of domesticated livestock, yet they will utilize the same grounds for grazing when livestock are absent [7]. This can be problematic for ranchers occupying areas near elk populations like the Greater Yellowstone Area. Elk are known reservoirs for brucellosis, (*Brucella abortus*) a disease that is highly contagious and poses a risk to livestock and humans [8–10]. Because of the potential for causing abortion in cattle, the USDA used vaccines and serologic testing to nearly eradicate *B. abortus* from domestic herds [11]. Yet in the last 15 years, over 20 cases of transmission to cattle have been traced to wild elk populations in the Greater Yellowstone Area. Attempts to establish long-term immunity through vaccination have proven unfruitful, as elk have negligible adaptive cellular immune responses to existing *Brucella* vaccines [12]. Because the eradication of *B. abortus* from cattle herds can cost hundreds of thousands of dollars and current tools make it unfeasible to control infection in wild elk, there is a need to dissect the genetic nature of limited immune responses in elk. With advances in sequencing technology (PacBio, Illumina and Hi-C), we are now able to investigate difference in adaptive immune response at the genomic level by examining the presence and absence of immune system-related genes. Here, we report a chromosomal level reference genome assembly and annotation of the Rocky Mountain elk and perform a preliminary investigation of immune gene loss between elk and cattle.

## Methods

### Animal selection

A long-captive herd in Minnesota provided a healthy adult male Rocky Mountain elk for Pac-Bio sequencing, and another for HiC and Chicago sequencing. White blood cells from six females from the aforementioned herd and six females from Wyoming were used for paired end sequencing, while an an elk calf, captive-born in Iowa, was used for RNA-seq. The research protocol was approved by the National Animal Disease Center Animal Care and Use committee and all animals under the protocol were maintained in accordance with animal care regulations.

### Sequencing

For the initial contig assembly we generated a hybrid data set with Illumina PCR-free 150bp paired end reads and PacBio RSII reads produced with P6-C4 chemistry. Chicago and Hi-C libraries were prepared as described previously [13, 14]. Both Chicago and Hi-C libraries were prepared similarly, though Hi-C libraries were nuclear-fixed. Briefly, formaldehyde-fixed chromatin was digested with *Dpn*II, and 5' overhangs were sealed with biotinylated nucleotides. Blunt ends were ligated, followed by crosslink were reversed for DNA purification from protein. We then removed biotin that was not internal to ligated fragments. DNA was sheared to a mean length of ~350 bp for library construction with NEBNext Ultra enzymes and Illumina-compatible adapters. Biotin-containing fragments were isolated using streptavidin beads before PCR enrichment of the libraries. Both Chicago and Hi-C libraries were sequenced on an Illumina HiSeqX at 2x150bp, attaining totals of 470 million and 500 million reads, respectively.

To prepare samples for PacBio and Illumina sequencing, DNA from purified peripheral blood mononuclear cells was isolated using a Gentra Puregene Blood Kit (Qiagen) and Genomic-tip 500/G kit (Qiagen), respectively, in accordance with manufacturer recommendations. Resulting DNA preparations were quantified using Qubit Broad Range Assay (ThermoFisher) and assessed for quality via Nanophotometer Pearl (Implen). Prior to Pacific Biosciences (PacBio) library preparation, DNA fragment size was evaluated using the HS Large Fragment 50 Kb method on fragment analyzer (Advanced Analytical Technologies,

Inc.) and determined to have an average size of approximately 40 kb. The DNA was sheared to approximately 20kb, size separated using a Blue Pippin using the PAC-30 KB cassette (Sage Science). Libraries were prepared for PacBio sequencing using the large insert library protocol and Illumina sequencing using the TruSeq PCR-free kit per manufacturer recommendations. Long read sequencing was conducted on the Pacific Biosystems RS II. Illumina short read sequencing (150 bp PE) was conducted on the HiSeq 3000 platform in accordance with manufacturer recommendations.

For preparation of RNAseq data tissue samples (skeletal muscle, spleen, kidney, lung, prescapular lymph node and mesenteric lymph node) were collected and stored in RNAlater™ (Ambion) at 4˚C. Excess RNAlater™ was removed following overnight incubation, and samples were stored at -80˚C. For RNA isolation, approximately 50 mg of each tissue were added to 1 ml of TRIzol© (ThermoFisher) and processed according to manufacturer's instructions. Following collection of the aqueous phase, samples were purified using the Purelink© RNA Mini kit (ThermoFisher), following manufacturer's recommendations. RNA quality was assessed using an Agilent Bioanalyzer using the RNA 6000 Nano kit. RNA concentrations were determined using a Nanodrop (ThermoFisher). Sequencing libraries were prepared after ribosomal RNA depletion using the Ribo-Zero H/M/R kit (Illumina) and stranded total RNA-seq libraries were prepared using the Ultra II RNA library prep kit (New England Biolabs) per manufacturer's recommendations. Resulting libraries were sequenced using a HiSeq 3000 (Illumina) and 100 cycle paired-end chemistries.

## Genome assembly

An initial genome assembly was generated with Masurca version 3.2.3 [15], attaining a 2,559.8 Mbp genome size in 29,125 contigs with N50 size of 1,224,689bp. Dovetail Genomics scaffolded this assembly using an iterative HiRise analysis informed via alignments of Chicago and then Hi-C libraries with a modified SNAP aligner (http://snap.cs.berkeley.edu). This assembly contained 2,560.5 Mb, with an L90 of 31 scaffolds, and a N90 of 43.374 Mb. 1,004,453,472 Chicago and Hi-C reads were used to scaffold this Dovetail assembly with a Juicer 1.5.6, 3D-DNA 180922, and JuiceBox 1.9.8 [16, 17]. Reads were extracted from bam files with Picard 2.9.2 [18]. The Dovetail assembly was masked using RepeatModeler 4.0.7 [19] and RepeatMasker 1.0.8 [20], prior to the alignment of Hi-C reads with BWA mem 0.7.17 [21]. Alignments were processed using Juicer, 3D-DNA [22], and Juicebox [16, 17]. The Juicebox assembly strategy consisted of: manually placing all contigs greater than 10kb, incorporating scaffolds at the highest Hi-C signal, placing scaffolds in non-repetitive regions when Hi-C signal was equal between a repetitive and non-repetitive region, repeats were clustered whenever possible, and only obvious mis-joins were edited. The initial Juicebox scaffolding created 34 pseudomolecules, which was then compared to the *Cervus elaphus hippelaphus* genome assembly (GCA_002197005.1) [23] to reveal the merger of the X and Y chromosomes. A BLASTn [24] of the *C. elaphus hippelaphus* genome sequence was used to identify coordinates, allowing the correct separation the X and Y chromosome via the heatmap in Juicebox. The 3D-DNA assembly finished with 22,557 scaffolds.

The contigs that could not be integrated into the pseudomolecules were eliminated based on repetitiveness, duplicated heterozygous contigs, RNA-seq mapping potential, and contig size (>500 bp). BEDTools 2.25.0 [25] was used to merge coordinates from mapping these contigs to the pseudomolecules with BLAST+ 2.9 (score >300) and RepeatMasker 1.0.8 [20] masking coordinates. 22,065 contigs were eliminated that were less than 1kb, had at least 90% query coverage, and lacked a single unique mapping RNA-seq read, leaving 35 pseudomolecules, 457 contigs, and a mitochondrial genome.

The assembly was polished with Pilon 1.23 [26] using CCS PacBio reads and paired end Illumina DNA-seq. CCS PacBio reads were created from the PacBio subreads using bax2bam [27] and Bamtools 2.5.1 [28] and then aligned using Minimap 2.6 [29]. Paired end reads were aligned using Hisat2 2.0.5 [30], followed by bam conversion and sorting with Samtools 1.9 [31]. Due to uneven and excessive coverage in repetitive regions, paired end alignments were set at a max coverage of 30x using jvarkit [32]. Due to the excessive repetitiveness of Chromosome_14, 50Mbp of this chromosome was not polished.

After polishing, another round of small contig elimination was performed by merging RepeatMasker [20] coordinates and coordinates from BLAST+ 2.9 [24] (score >300, width 1000bp) to the pseudomolecules with Bedtools 2.25.0 [25]. If 90% of query length was repetitive and contained within the pseudomolecules, it was eliminated. BlobTools 1.11 [33] was run with PacBio subread alignments to the genome, and contigs annotated with BLAST [24] to the NT database (S1 Fig). All scaffolds passed contamination screening, resulting in a final assembly containing 35 pseudomolecules, 151 contigs, and the mitochondrion.

## Mitochondrial identification and annotation

BLAST+ 2.9 [24] was used to identify the mitochondrial genome by querying the mitochondrial scaffold of the *C. elaphus hippelaphus* GCA_002197005.1 [23]. Though the mitochondrial genome was identified, it contained three juxtaposed mitochondrial genome duplications. The scaffold was manually corrected using genomic coordinates with faidx in Samtools 1.9 [31]. Genes were annotated in the mitochondrial genome using the Mitos2 webserver [34] with RefSeq 89 Metazoa, a genetic code of 2, and default settings.

## Repeat prediction

A final version of predicted repeats was obtained using–sensitive 1 and–anno 1 for EDTA 1.7.9 [35] and with default parameters for RepeatModeler 1.0.8 [19] with RepeatMasker 4.1.0 [20].

## Gene prediction

A total of 753,228,475 RNA-seq reads aligned to the genome using Hisat2 2.0.5 [30] followed by bam conversion and sorting with Samtools 1.9 [31]. RNA-seq read counts were obtained using Subread 1.5.2 [36]. The alignments were assembled into genome-guided transcriptomes using Trinity 2.8.4 [37–39], Strawberry 1.1.1 [40], Stringtie 1.3.3b [41, 42], and Class2 2.1.7 [43]. The RNA-seq alignments were also used for a gene prediction via Braker2 2.1.4 [44] with Augustus 3.3.3 [45] on a genome soft-masked by RepeatMasker 1.0.8 [20] with a custom RepeatModeler 4.0.7 [19] library. High confidence exon splicing junctions were identified using Portcullis 1.1.2 [46]. Each of these assemblies were then supplied to Mikado 2.0rc6 [47] to pick consensus transcripts, while utilizing Cervus-specific proteins from Uniprot [48] (downloaded 12-28-19). This mikado prediction was filtered for transposable elements using Bedtools 2.25.0 intersect [25] and filtered for pseudogenes via removing genes with five or fewer mapping RNA-seq reads. With Bedtools 2.25.0 [25] intersect these filtered Mikado gene models were used to find corresponding Braker2 2.1.4 [44] gene models. Both of these predictions, together with a Genomethreader 1.7.1 [49] alignment of Uniprot proteins from the Pecora infraorder (downloaded 02-07-20) were used for a final round of Mikado gene prediction. The predicted transcripts and proteins were generated using Cufflinks [50] gffread (2.2.1), and subjected to functional annotation to: Interproscan 5.27–66.0 [51, 52] and BLAST [24] searches to NCBI NT and NR databases downloaded on 10-23-19, as well as Swissprot/ Uniprot databases downloaded on 12/09/2019.

## BUSCO

Universal single copy orthologs were assessed using BUSCO 4.0 [53, 54], with the eukaryota_odb10 and cetartiodactyla_odb10 datasets in both genome and protein mode.

## Synteny

With the predicted proteins from *B. taurus* (GCF_002263795.1_ARS-UCD1.2) [55], *C. elaphus* (GCA_002197005.1) [23] and *C. canadensis* genome assemblies, we inferred gene orthology using BLASTp [24], at cutoffs of an e-value of 1e-5, 50% query cover, and 70% identity. Gene-based synteny was predicted using iAdHoRe 3.0.01 [56] with prob_cutoff = 0.001, level 2 multiplicons only, gap_size = 5, cluster_gap = 15, q_value = 0.01, and a minimum of 3 anchor points. Synteny figures were produced using Circos (0.69.2) [57]. Dot plots were produced using MCScanX 20170403 [58].

## Identification and verification of immune system-related genes

Immune system-related genes from *Bos taurus* were found in the GENE-DB database of the International ImMunoGeneTics website (www.imgt.org) [59]. This database is comprised of immunoglobulins (IG), T cell receptors (TR) and major histocompatibility (MH) genes from vertebrate species. A tblastn (2.9.0+) [24] was performed against the elk and cattle genome assembiles (GCF_002263795.1_ARS-UCD1.2) [55], with an e-value cutoff of 1e-3. We removed candidate missing genes based on whether a similar isoform was present in the elk genome. To continue finding candidate missing genes in the elk genome, not found by tBLASTn, we investigated using Bedtools 2.25.0 extracted cattle nuceotide sequences with a BLASTn to the elk genome. Those genes that were still not found via BLASTn [24], were modified to retain 20 bp border sequences with Bedtools 2.25.0, and subjected to another BLASTn [24] to the elk genome. If a gene was still not found, hit sequences in the cattle genome were expanded by 100bp with Bedtools 2.25.0, combined with the elk genome, and used for Hisat2 2.0.5 [30] RNAseq mapping and Minimap2 2.6 [29] Pacbio mapping. Read counts were discerned using FeatureCounts from the Subread package 1.5.2 [36].

# Results and discussion

Here we present the first pseudomolecule assembly of *C. canadensis*, generated with 1.7 trillion base pairs of sequencing at a 686-fold coverage of the genome.

## Genome assembly

An initial assembly was created with MaSuRCA [15, 60] generating 23,302 contigs, an L90 of 2,500 contigs, and an N90 of 197,963bp. Through collaboration with Dovetail Genomics and then additional implementation of the Juicer/JuiceBox/3D-DNA pipeline [16, 17, 22], we generated an assembly of 33 autosomes, an X chromosome, a Y chromosome, a mitochondrial genome, and 151 unincorporated contigs. This result is supported by published cytological studies revealing a haploid set of 34 chromosomes [61]. We utilized synteny to identify homologous chromosomes between elk and red deer, and found that nearly always, elk chromosome sizes fell within the estimated size of the red deer's assembled chromosomes [23] (S1 Table). The only exception is the Y chromosome, which was nearly twice (7.6 Mb) the largest predicted size (4 Mb) of the red deer chromosome. We investigated all putative contaminant contigs from Blobtools [33], and ruled out contamination (S1 Fig), but also took additional steps to ensure the completeness of the genome by mapping reads back to the assembly. We found that we captured the majority of genome, with 90.7% and 87.3% of PacBio CCS reads Illumina

DNA-seq aligning to the genome (S2 Table). To evaluate the completeness of the genome we ran BUSCO 4.0.2 [54] (Benchmarking Universal Single Copy Orthologs) on genome. Of the possible 255 and 13,335 genes in the eukaryota and certartiodactyla odb10 datasets, 62% and 88.1% were complete, 2.4% and 2.1% were duplicated, and 3.1% and 2.1% were fragmented, and 32.5% and 9.8% were missing, respectively.

## Genome annotation

To obtain a high-quality elk gene prediction, we pursued an extensive annotation of repeats in the genome using two repeat predictors. While EDTA [35] utilizes a comprehensive set of repeat prediction programs to create a repeat annotation, Repeatmodeler/Repeatmasker [19, 20] is a long-standing and comparable annotator of repeats that is more reliant on copy number. With EDTA, 25.8% of the genome was marked repetitive, with DNA transposons comprised the largest percentage of repeats in the genome, at 16% (S3 Table). In contrast, RepeatMasker assessed 36.5% of the genome as an interspersed repeat, with 28.8% of the genome being comprised LINE retrotransposons. We merged these repeat annotations with BEDTools [25] to reveal that 38% of the genome is repetitive. This is in contrast to the repetitive content in red deer, estimated at 22.7%. This difference could be due to technological improvements and could stem from the large proportion of gaps in the red deer genome (1.5Gbp) [23]. While together these differences could account for a large disparity in chromosome sizes, only the elk Y chromosome was outside the gapped and sequence length range in red deer chromosomes [23].

To annotate the genes in the genome we generated 1.5 billion paired end reads of sequencing from six tissues, including kidney, lung, mesenteric lymph node, muscle, prescapular lymph node, and spleen. After masking repeat sequences using Repeatmodeler [19] and Repeatmaker [20], we performed five de novo transcript/gene predictions with a soft-masked genome and RNA-seq. The best transcripts were discerned using Mikado [47], followed by clustering with Cufflinks [50] using *B. taurus* mRNAs to cluster transcripts into gene loci. Using this approach 18,013 genes were predicted to encode 33,433 mRNAs (S4 Table). The functional annotations of these genes were extremely high, with 17,938 of the 18,013 genes or 99.6% being annotated by at least one of: Interproscan or BLAST to NR, NT, and Uniprot (S5 Table). The gene annotation was evaluated for completeness with BUSCO in protein mode. A remarkable "Complete" score improvement is seen in both eukaryota and cetartiodactyla at 97.7% and 92.1%, respectively. These results together suggest that both the genome and the gene prediction are of high quality.

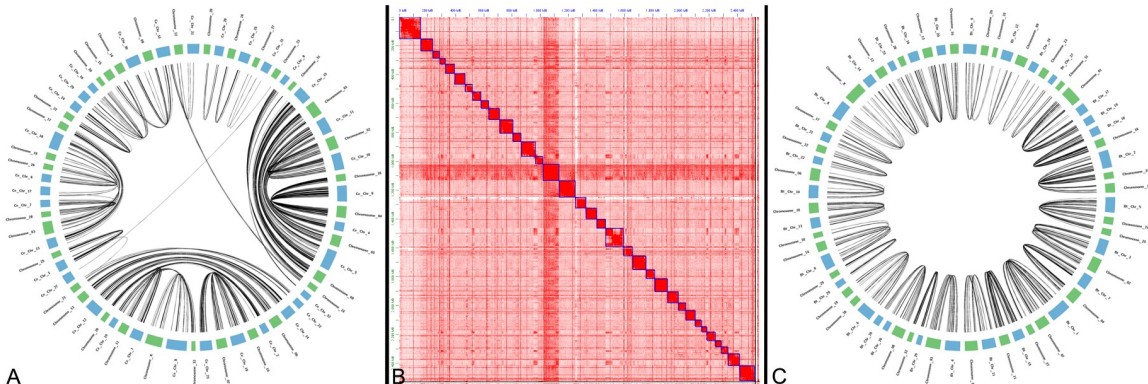

**Fig 1. Synteny and Hi-C plot of elk chromosomes.** A. Gene-based synteny between *C. elaphus hippelaphus* and *C. canadensis*. B. Hi-C plot of elk chromosomes in JuiceBox. C. Gene-based synteny between *B. taurus* and *C. canadensis*.

## Comparison to related species

By utilizing these new gene predictions we evaluated the conservation of chromosome structure between *C. canadensis*, *C. elaphus hippelaphus*, and *B. taurus* using gene-based synteny with i-ADHoRe [56]. All elk chromosomes were syntenic with all *C. elaphus* and *B. taurus* chromosomes, though Y chromosome lacked the genes required for gene-based synteny (Fig 1, Table 1). As has been seen in previous Cervus assemblies [23], multiple pairs of chromosomes are tandemly fused in *B. taurus* and vise-versa (Table 2). We confirmed previous reports

**Table 1. Chromosome statistics of the Rocky Mountain elk assembly compared to red deer, with syntenic relationships to red deer, sika deer, cattle, sheep and human.**

| *Cervus canadensis* | Total length (bp) | Repetitive elements (bp) | Gene Frequency | Red Deer Gene Frequency | Chromosomal Relationships | | | | |
|---|---|---|---|---|---|---|---|---|---|
| | | | | | Red deer | Sika deer | Cattle | Sheep | Human |
| 1 | 127,605,827 | 46,694,602 | 1,460 | 1,698 | 5 | 2 | 17, 19 | 17, 11 | 4, 12, 17 |
| 2 | 114,865,875 | 43,848,496 | 999 | 1,132 | 20 | 3 | 3 | 1 | 1 |
| 3 | 114,606,702 | 42,403,479 | 631 | 626 | 18 | 4 | 4 | 4 | 7 |
| 4 | 105,318,381 | 40,480,415 | 925 | 1,025 | 9 | 5 | 7 | 5 | 5, 19 |
| 5 | 101,869,976 | 36,732,257 | 864 | 910 | 11 | 8 | 11 | 3 | 2, 9 |
| 6 | 96,780,817 | 34,856,794 | 718 | 794 | 12 | 16 | 10 | 7 | 14, 15 |
| 7 | 94,470,602 | 36,360,279 | 554 | 619 | 19 | 7 | 1 | 1 | 3, 21 |
| 8 | 92,076,199 | 33,431,109 | 602 | 712 | 15 | 9 | 26, 28 | 22, 25 | 1, 10 |
| 9 | 84,228,583 | 32,593,999 | 358 | 382 | 30 | 10 | 12 | 10 | 13 |
| 10 | 82,287,371 | 29,138,716 | 705 | 687 | 23 | 1q | 13 | 13 | 10, 20 |
| 11 | 78,153,912 | 31,079,399 | 603 | 622 | 1 | 11 | 15 | 15 | 11 |
| 12 | 77,654,944 | 28,351,493 | 432 | 409 | 21 | 13 | 14 | 9 | 8 |
| 13 | 76,089,960 | 28,668,740 | 563 | 587 | 14 | 14 | 16 | 12 | 1 |
| 14 | 74,494,459 | 26,159,099 | 320 | 307 | 29 | 15 | 8 | 2 | 9 |
| 15 | 74,380,151 | 29,044,063 | 280 | 463 | 33 | 12 | 2, 22 | 2 | 2, 3 |
| 16 | 67,981,682 | 25,953,664 | 304 | 289 | 25 | 20 | 20 | 16 | 5 |
| 17 | 65,378,136 | 25,514,684 | 475 | 472 | 13 | 21 | 21 | 18 | 14, 15 |
| 18 | 64,413,554 | 22,951,146 | 971 | 1,035 | 4 | 1p | 18 | 14 | 19 |
| 19 | 62,010,818 | 24,221,065 | 204 | 246 | 17 | 16 | 6 | 6 | 4 |
| 20 | 60,444,953 | 24,378,692 | 215 | 245 | 28 | 17 | 9 | 8 | 6, 9 |
| 21 | 59,747,184 | 22,203,178 | 560 | 520 | 22 | 19 | 5 | 3 | 22 |
| 22 | 59,530,028 | 20,562,536 | 498 | 519 | 24 | 26 | 22 | 19 | 3 |
| 23 | 58,383,784 | 20,478,363 | 276 | 321 | 27 | 24 | 24 | 23 | 18 |
| 24 | 54,121,439 | 19,309,984 | 480 | 455 | 8 | 18 | 2 | 2 | 1, 2 |
| 25 | 53,619,048 | 20,223,354 | 382 | 530 | 3 | 27 | 5 | 3 | 12 |
| 26 | 52,893,355 | 19,063,751 | 287 | 333 | 6 | 22 | 6 | 6 | 4 |
| 27 | 52,039,427 | 21,233,487 | 164 | 193 | 31 | 25 | 1 | 1 | 21 |
| 28 | 51,438,166 | 17,786,547 | 534 | 492 | 7 | 23 | 23 | 20 | 6 |
| 29 | 48,396,561 | 18,012,957 | 521 | 541 | 2 | 29 | 29 | 21 | 11 |
| 30 | 44,123,562 | 16,926,467 | 302 | 327 | 16 | 32 | 8 | 2 | 8, 9 |
| 31 | 42,799,129 | 15,135,670 | 211 | 196 | 32 | 28 | 27 | 26 | 4, 8 |
| 32 | 40,102,283 | 14,331,760 | 611 | 702 | 10 | 30 | 25 | 24 | 7, 16 |
| 33 | 38,432,887 | 12,811,166 | 223 | 240 | 26 | 31 | 9 | 8 | 6 |
| X | 146,388,637 | 74,117,965 | 744 | 716 | X | X | X | X | X |
| Y | 7,618,728 | 4,865,392 | 27 | 23 | Y | Y | Y | Y | Y |
| Unplaced | 1,865,887 | 19,491 | 10 | 10 | | | | | |
| Total | 2,526,613,007 | 959,944,259 | 18,013 | 19,378 | | | | | |

**Table 2. Chromosomal fissions and fusions between elk and cattle genomes.**

| C. canadensis | B. taurus |
|---|---|
| 25,21 | 5 |
| 19,26 | 6 |
| 14,30 | 8 |
| 20,33 | 9 |
| 24,15 | 2 |
| 7,27 | 1 |
| 1 | 17,19 |
| 8 | 26,28 |

of chromosome fusions and fissions indicated that twelve cervus chromosomes fused into six in *B. taurus*, as well as four chromosomes in *B. taurus* are fused into two cervus chromosomes (Table 2).

Two inter-chromosomal translocations were inferred between the two Cervus species, both having strong Hi-C support in elk (Fig 1, Table 3). Chromosome_15 and Chromosome_24 of elk, comprised large portions of *C. elaphus* Ce_Chr_33 and a minor portion of Ce_Chr_8. With the majority of Chromosome_24 homologous to *C. elaphus hippelaphus* Ce_Chr_8, a 17 MB region of Ce_Chr_33 may have been falsely attached to Ce_Chr_8 in *C. elaphus hippelaphus*. Another smaller chromosome translocation of 13.6 MB occurred between Ce_Chr_22 and Ce_Chr_3 of C. elaphus, attributed to chromosomes 21 and 25 in *C. canadensis*. A small region of Ce_Chr_22 was likely falsely attached to Ce_Chr_3 in *C. elaphus hippelaphus*. Interestingly, both of these translocations are between chromosomes in elk that are fused chromosomes in *B. taurus*, Bt_Chr_2 and Bt_Chr_5 (Table 3). While it is possible that these translocations occurred since the divergence of these two species, because the *B. taurus* assembly was used to orient and join scaffolds in the *C. elaphus hippelaphus* genome assembly, it is likely that these translocations are misassemblies in the *C. elaphus hippelaphus* genome.

## Immune gene loss

A total of 36 *Bos taurus* immune coding sequences from the IMGT GENE-DB database [59] were lacking from initial investigations of the elk genome, and yet were identified in cattle genome. Despite extensive attempts to identify these genes in the elk genome with tBLASTn, BLASTn of cattle hit sequences, and BLASTn of cattle hit sequences with 20bp borders, we were unable to identify putative elk orthologs (Table 4, S6 Table). However, seventeen putative gene loci were identified in elk using a BLASTn of cattle nucleotide sequences hit by the tBLASTn, an additional twelve were found using the broadened cattle hit sequences with 20bp borders, and seven were confirmed missing from the genome (S6 Table, Table 4). We found a

**Table 3. Inter-chromosomal translocation comparisons among Cervus species and cattle.**

| C. canadensis | C. elaphus | B. taurus |
|---|---|---|
| 15 | 33,8 | 2p |
| 24 | 8 | 2q |
| 21 | 22,3 | 5p |
| 25 | 3 | 5q |

Ce_Chr_8 has a 17Mbp region of Ce_Chr_33, and Ce_Chr_3 has a 13.6Mb region of Ce_Chr_22. P is proximal, q represents distal.

**Table 4. Read mapping of suspected missing genes in the elk genome.**

| | Gene Name | kidney_S25_L003 | kidney_S25_L004 | lung_S26_L003 | lung_S26_L004 | Mes-LN_S24_L003 | Mes-LN_S24_L004 | muscle_S21_L003 | muscle_S21_L004 | pscapLN_S22_L003 | pscapLN_S22_L004 | spleen_S23_L003 | spleen_S23_L004 | PacBio |
|---|---|---|---|---|---|---|---|---|---|---|---|---|---|---|
| **Blastn Only** | D13648_TRGJ3-1 | 0 | 1 | 1 | 1 | 24 | 15 | 0 | 0 | 21 | 18 | 37 | 28 | 0 |
| | AY644517_TRGC3 | 0 | 0 | 5 | 3 | 16 | 22 | 0 | 0 | 31 | 25 | 50 | 39 | 0 |
| | AY644517_TRGC4 | 0 | 0 | 0 | 0 | 0 | 1 | 0 | 0 | 2 | 1 | 1 | 0 | 0 |
| | IMGT000049_TRAJ2 | 3 | 4 | 10 | 8 | 129 | 117 | 2 | 0 | 31 | 27 | 18 | 19 | 0 |
| | IMGT000049_TRAJ5 | 1 | 3 | 9 | 15 | 94 | 97 | 1 | 0 | 21 | 20 | 9 | 13 | 0 |
| | IMGT000049_TRAJ8-1 | 0 | 0 | 0 | 0 | 0 | 0 | 0 | 0 | 0 | 0 | 0 | 0 | 1 |
| | IMGT000049_TRAJ8-1 | 0 | 0 | 0 | 0 | 0 | 0 | 0 | 0 | 0 | 0 | 0 | 0 | 0 |
| | IMGT000049_TRAJ19 | 3 | 2 | 8 | 6 | 143 | 117 | 0 | 2 | 13 | 26 | 15 | 10 | 0 |
| | AY227782_TRAJ25 | 4 | 4 | 8 | 6 | 143 | 138 | 0 | 0 | 24 | 20 | 14 | 11 | 1 |
| | IMGT000049_TRAJ29 | 3 | 4 | 5 | 12 | 122 | 119 | 0 | 0 | 20 | 19 | 16 | 15 | 0 |
| | AY227782_TRAJ31 | 0 | 2 | 5 | 4 | 67 | 77 | 0 | 0 | 16 | 12 | 4 | 8 | 0 |
| | IMGT000049_TRAJ34 | 1 | 3 | 11 | 8 | 123 | 108 | 0 | 1 | 20 | 23 | 12 | 11 | 1 |
| | IMGT000049_TRAJ35 | 5 | 7 | 6 | 7 | 108 | 129 | 0 | 0 | 36 | 21 | 15 | 7 | 0 |
| | IMGT000049_TRAJ38 | 3 | 3 | 3 | 5 | 84 | 102 | 0 | 0 | 19 | 21 | 22 | 8 | 2 |
| | IMGT000049_TRAJ48 | 1 | 3 | 3 | 7 | 91 | 68 | 0 | 0 | 15 | 14 | 11 | 8 | 0 |
| | IMGT000049_TRAJ57 | 2 | 3 | 1 | 2 | 26 | 16 | 0 | 0 | 3 | 7 | 5 | 3 | 0 |
| | KT723008_IGHD1-3 | 1 | 1 | 4 | 5 | 110 | 128 | 0 | 0 | 192 | 14 | 173 | 16 | 1 |
| **Blastn +20bp borders** | AC172685_TRGJ2-1, D16118_TRGJ2-1 | 0 | 0 | 0 | 0 | 11 | 3 | 0 | 0 | 14 | 20 | 27 | 24 | 1 |
| | IMGT000049_TRAJ6 | 2 | 6 | 8 | 7 | 121 | 118 | 2 | 0 | 29 | 50 | 18 | 11 | 0 |
| | IMGT000049_TRAJ8-2 | 1 | 0 | 2 | 1 | 20 | 22 | 0 | 0 | 4 | 2 | 3 | 2 | 0 |
| | IMGT000049_TRAJ8-2 | 1 | 0 | 0 | 1 | 15 | 20 | 0 | 0 | 1 | 7 | 0 | 3 | 1 |
| | IMGT000049_TRAJ11 | 4 | 3 | 13 | 14 | 194 | 198 | 2 | 0 | 26 | 37 | 21 | 25 | 0 |
| | IMGT000049_TRAJ12 | 6 | 8 | 5 | 7 | 142 | 167 | 1 | 0 | 31 | 21 | 23 | 15 | 0 |
| | IMGT000049_TRAJ27 | 3 | 3 | 6 | 5 | 191 | 155 | 0 | 0 | 27 | 35 | 27 | 23 | 0 |
| | IMGT000049_TRAJ33 | 5 | 5 | 7 | 8 | 114 | 119 | 0 | 0 | 26 | 31 | 18 | 16 | 0 |
| | IMGT000049_TRAJ40 | 0 | 2 | 12 | 6 | 103 | 100 | 0 | 0 | 22 | 16 | 12 | 16 | 1 |
| | IMGT000049_TRAJ46 | 0 | 6 | 6 | 2 | 116 | 132 | 1 | 1 | 21 | 19 | 7 | 8 | 0 |
| | IMGT000049_TRDC | 8 | 5 | 83 | 89 | 133 | 112 | 1 | 0 | 298 | 285 | 192 | 212 | 0 |
| | KT723008_IGHJ2-1 | 0 | 0 | 0 | 1 | 25 | 44 | 0 | 0 | 12 | 10 | 4 | 14 | 1 |
| **Not Found** | IMGT000049_TRAJ3 | 0 | 0 | 0 | 0 | 0 | 0 | 0 | 0 | 0 | 0 | 0 | 0 | 0 |
| | IMGT000049_TRAJ17 | 0 | 0 | 0 | 0 | 0 | 0 | 0 | 0 | 0 | 0 | 0 | 0 | 0 |
| | IMGT000049_TRAJ42 | 0 | 0 | 0 | 0 | 0 | 0 | 0 | 0 | 0 | 0 | 0 | 0 | 0 |
| | IMGT000049_TRAJ49 | 0 | 0 | 0 | 0 | 0 | 0 | 0 | 0 | 0 | 0 | 0 | 0 | 0 |
| | IMGT000049_TRAJ56 | 0 | 0 | 0 | 0 | 0 | 0 | 0 | 0 | 0 | 0 | 0 | 0 | 0 |
| | KT723008_IGHD | 0 | 0 | 0 | 0 | 0 | 0 | 0 | 0 | 0 | 0 | 0 | 0 | 0 |
| | AY14283_IGHJ1-2, KT723008_IGHJ2-2, NW_001494075_IGHJ1-2 | 0 | 0 | 0 | 0 | 0 | 0 | 0 | 0 | 0 | 0 | 0 | 0 | 0 |

Tissues assessed were kidney, lung, mesenteric lymph node, muscle, pre-scapular lymph node, and spleen. Blastn are those genes only found with BLASTn of cattle tBLASTn hit sequences. Blastn +20bp are only those genes found by including 20bp surrounding the cattle tBLASTn hit sequences. Not Found are those genes that did not have homology to the genome nor the transcriptomic/genomic data.

complete lack of genomic gaps in these regions, confirming the contiguity of these suspected gene regions. However, RNA-seq aligned to 27/36 of these suspected loci, indicating genomic variation in these regions may prevent their identification. Nevertheless, nine genes lacked a translatable sequence in the elk genome and could not align RNAseq, confirming their absence from both genomic and transcriptomic data. These genes were AY644517_TRGC4, IMGT000049_TRAJ8-1, IMGT000049_TRAJ3, IMGT000049_TRAJ17, IMGT000049_TRAJ42, IMGT000049_TRAJ49, IMGT000049_TRAJ56, KT723008_IGHD, and a homolog of (AY149283_IGHJ1-2,KT723008_IGHJ2-2,NW_001494075_IGHJ1-2) (S6 Table). All of these loci encode components of the T cell receptor: (gamma constant 2), (T cell receptor alpha joining), and (delta chain) or are heavy chains in the immunoglobulin complex (S6 Table).

Ruminants, including elk, differ from rodents and humans by the high proportion (sometimes 40–50%) of T cells circulating in the peripheral blood expressing γδ receptors. In all species, γδ T cells are involved in diverse and important roles in not only adaptive, but also innate immune responses [62]. Rearrangements of V (variable), J (joining) and C (constant) regions of the γ chain when combined with the δ chain contribute to the repertoire diversity of the γδ T cell receptor. While future work will be necessary to understand how the loss of these genes affects the cellular immune response in elk, certainly the loss of T-cell receptor diversity is an important consideration in discerning why elk does not develop protective immunity after *B. abortus* vaccination. Because *B. abortus* is a facultatively intracellular bacteria, stages of the disease cannot be accessed by antibodies, and thus cellular immune responses must be activated by T cell receptors interacting with antigens on the surface of infected cells [63, 64]. In cattle, protection to some bacterial diseases via vaccines is mediated by memory T cells activating effector T cells and some specific cases, effector T cell populations bearing gamma-delta chain receptors. A reduction in the number of available T cell receptor variants could limit or hinder immune responses to some antigens. Thus, this investigation provides a foundation for the development of a viable vaccination strategy in elk, a step towards developing long-term immunity to *Brucella*.

## Conclusions

This genome assembly and annotation of the Rocky Mountain elk is the most contiguous assembly of a Cervus species and will serve as an important tool for genomic exploration of all related Cervids. Elk's loss of immune system-related genes in relation to cattle, may provide a clue to establishing a successful vaccination strategy. This chromosomal assembly of the elk genome will provide an excellent resource for investigating genes involved in elk's poor adaptive cellular immune response to Brucella vaccines.

## Supporting information

**S1 Table. Chromosomal lengths and syntenic relationships between *C. canadensis* and *C. elaphus hippelaphus*.**
(XLSX)

**S2 Table. Mapping of reads used in assembly and annotation.**
(XLSX)

**S3 Table. Repeat predictions on the *C. canadensis* genome with EDTA and RepeatModeler with RepeatMasker.** The total is the overlapping content of these two annotations.
(XLSX)

**S4 Table. Statistics of genes, transcripts, and exons for all intermediate annotations used for the final annotation.**
(XLSX)

**S5 Table. Genes and mRNAs annotated by various databases for function.**
(XLSX)

**S6 Table. Annotations of putative missing immune gene loci.**
(XLSX)

**S1 Fig.**
(TIF)

**S1 File.**
(DOCX)

## Acknowledgments

The authors would like to thank Maryam Sayadi for fruitful discussion regarding the genome assembly paper, Mary Wood regarding elk sample collection, and the ISU DNA Sequencing facility for preparation of libraries and DNA sequencing. The Ceres cluster (part of the USDA SCInet Initiative) was used for computational resources.

## Author Contributions

**Conceptualization:** Rick E. Masonbrink, Aleksey Zimin, Steven Olsen.

**Data curation:** Rick E. Masonbrink.

**Formal analysis:** Rick E. Masonbrink, Aleksey Zimin, Andrew Severin.

**Funding acquisition:** Aleksey Zimin, Andrew Severin, Steven Olsen.

**Investigation:** Rick E. Masonbrink, David Alt, Darrell O. Bayles, Paola Boggiatto, William Edwards, Fred Tatum, Jeffrey Williams, Jennifer Wilson-Welder, Aleksey Zimin, Andrew Severin.

**Methodology:** Rick E. Masonbrink, Aleksey Zimin, Andrew Severin.

**Project administration:** Andrew Severin, Steven Olsen.

**Resources:** Rick E. Masonbrink, Aleksey Zimin.

**Software:** Rick E. Masonbrink.

**Supervision:** Steven Olsen.

**Validation:** Rick E. Masonbrink.

**Visualization:** Rick E. Masonbrink.

**Writing – original draft:** Rick E. Masonbrink.

**Writing – review & editing:** Rick E. Masonbrink, David Alt, William Edwards, Jennifer Wilson-Welder, Aleksey Zimin, Andrew Severin, Steven Olsen.

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
