## [Decision Letter · Decision Letter 0]

11 Jan 2021

PONE-D-20-33251

A pseudomolecule assembly of the Rocky Mountain elk genome reveals putative immune system gene loss near chromosomal fissions

PLOS ONE

Dear Dr. Masonbrink,

Thank you for submitting your manuscript to PLOS ONE. After careful consideration, we feel that it has merit but does not fully meet PLOS ONE’s publication criteria as it currently stands. Therefore, we invite you to submit a revised version of the manuscript that addresses the points raised during the review process.

The expert reviewers have provided valuable feedback that will improve the manuscript .  Please address each and every point.

We look forward to receiving your revised manuscript.

Kind regards,

F. Alex Feltus, Ph.D.

Academic Editor

PLOS ONE

Journal Requirements:

2. As part of your revision, please complete and submit a copy of the Full ARRIVE 2.0 Guidelines checklist, a document that aims to improve experimental reporting and reproducibility of animal studies for purposes of post-publication data analysis and reproducibility: https://arriveguidelines.org/sites/arrive/files/Author%20Checklist%20-%20Full.pdf (PDF). Please include your completed checklist as a Supporting Information file. Note that if your paper is accepted for publication, this checklist will be published as part of your article.

3. Please amend the manuscript submission data (via Edit Submission) to include author Jenny Wilson-Welder.

4. Please include a copy of Tables 1, 2, 3 and 4 which you refer to in your text on page 9 and 10.

Reviewers' comments:

Reviewer's Responses to Questions

**Comments to the Author**

1. Is the manuscript technically sound, and do the data support the conclusions?

Reviewer #1: Yes

Reviewer #2: Partly

2. Has the statistical analysis been performed appropriately and rigorously? 

Reviewer #1: N/A

Reviewer #2: N/A

3. Have the authors made all data underlying the findings in their manuscript fully available?

Reviewer #1: No

Reviewer #2: Yes

4. Is the manuscript presented in an intelligible fashion and written in standard English?

Reviewer #1: Yes

Reviewer #2: Yes

5. Review Comments to the Author

Reviewer #1: ## General comments ##

The authors have provided a genome assembly for Rocky Mountain elk (Cervus canadensis), which is a resevoir for Brucella abortus (a bacterium causing abortion). The genome assembly seems to be well put together and so is the annotation (see below for a couple of questions). They compare the immune gene complement of cattle to the elk genome, and find that elk is likely missing several immune genes.

The authors basically indicate that these missing genes might be the cause of the poor respons to vaccination in elk. It would have been good to see more discussion around this topic when you do raise it. Do you actually pin-point the genetic basis for this inherent difference, or did you just find some missing genes that happened to be immune genes (and no relation to vaccination)?

## Specific comments ##

Line 57: Comma should be after the reference.

Genome assembly:

It is common to polish with PacBio reads using arrow (https://github.com/PacificBiosciences/GenomicConsensus). Why did you not do this? If you had done this on the scaffolds/pseudochromosomes, you might have closed some smaller gaps even.

Line 115: How much coverage did you have with CCS reads? It is not clear to me (maybe I missed it) exactly what kind of PacBio library you created. I assume you tried to get the longest reads possible (please add information about this to the manuscript), and then you’d likely not have much coverage in CCS reads.

Line 125: Or do you mean “the mitochondrion»?

Line 129: How do you correct the scaffold with samtools? Just samtools faidx and coordinates?

Line 148: Do you mean that you ran this through Mikado once more? Interesting approach that results in quite conservative gene annotation I would assume.

Line 182-3: You didn’t map the CLRs to the genome? They would likely map at a lower rate, but could be interesting to see.

Line 233: Loss of genes is always a bit iffy to discuss. For instance, while you did not find these genes in the genome, were they also lacking from the reads? (Or from the transcriptome reads.) Or, did you look at the (micro)synteny in the affected regions and saw that where you expected the genes to be, there were none and no gaps or otherwise suspiscious sequence? I would like either of these investigations to be done. So, either confirm that the genes are also lacking from the raw reads, or confirm that they are not found based on synteny.

Line 246: How would the lack of these genes be utilized in developing a vaccine? I don’t know myself, but when you state this, I would like a bit more elaboration. One approach would be to get transcription data from infected/non-infected individuals to see what the immune system actually does.

Line 258: The github site for all programs and scripts were not available when I tried accessing it (27th November). It is great that you provide the scripts and such in that way, but unfortunate that I could not go in and browse the repository. The SRA project is also not available (PRJNA657053).

Reviewer #2: The Masonbrink et al. work describes a chromosome-level genome assembly for the Rocky Mountain elk. This new resource will be extremely valuable for the understanding of the elk's immune system and consequently to the prevention of the spread of brucellosis from elk to cattle.

The manuscript is well written and is easy to follow and the putative loss of immune-related genes in the elk show promise for better understand the differences between elk and cattle immune system.

I do have a few comments that I hope will help further improve it.

My main comment relates to the identification of putative gene losses. The result is very interesting and promising. Nonetheless, I believe this section needs a couple of extra checks to support these findings, as these are highlighted in the title and will set this work apart from a regular genome assembly report. For example, could any DNA or RNA seq reads be mapped to the cattle gene sequence? Since many of the putative missing genes are at the end of cattle chromosomes they could be within repetitive regions that are more difficult to assemble, but these sequences could still be recovered in the raw data. Other tool the authors could consider using is TOGA (https://github.com/hillerlab/TOGA) which uses pairwise genome alignments to infer orthologous genes between related species and to accurately distinguish orthologs from paralogs or processed pseudogenes. This could be useful to identify genes that were completely lost from pseudogenes, and also identify the mutations which could have led to pseudogenization.

Other comments:

1) Page 6 line 132: the parameters used for repeat annotation are not indicated.

2) Page 6 line 135: RNA-seq library prep and tissues/cells used are not mentioned in the methods section. The tissues used are listed in the results (page 10 line 201), but I believe they should also be indicated in this section.

3) Page 8 line 175: is the number of assembled molecules the expected number of chromosomes of the elk? Maybe a citation to a cytogenetics work with that information could be added (e.g. Koulischer, L., et al. (1972). Mammalian cytogenetics. VII. The chromosomes of Cervus canadensis, Elaphurus davidianus, Cervus nippon (Temminck) and Pudu pudu. Acta Zoologica et Pathologica Antverpiensia, 56, 25-30).

4) Page 8 line 185: is there a reason behind not using the mammalian busco set? If used it might give the opportunity to compare the completeness of the assembled genome to that of C. elaphus.

5) The reviewer pack did not contain supplemental tables so I could not review those.

6) There are a few typos or multiple versions of the same word across the manuscript that should be fixed for consistency (e.g. Hi-C vs HI-C vs HiC; elk vs Elk; missing spaces after references; extra commas).

6. PLOS authors have the option to publish the peer review history of their article (what does this mean?). If published, this will include your full peer review and any attached files.

Reviewer #1: **Yes: **Ole K. Tørresen

Reviewer #2: **Yes: **Joana Damas

---

## [Author Response · Author response to Decision Letter 0]

9 Feb 2021

Dear Reviewers,

Thank you for giving your time to review this manuscript. Your reviews have provided the context we needed to make a clearer story, which we appreciate greatly. We hope that we have addressed all of the concerns and look forward to your thoughts. Please see addressed specific comments below. 

Rick Masonbrink

Reviewer #1: ## General comments ##

The authors have provided a genome assembly for Rocky Mountain elk (Cervus canadensis), which is a reservoir for Brucella abortus (a bacterium causing abortion). The genome assembly seems to be well put together and so is the annotation (see below for a couple of questions). They compare the immune gene complement of cattle to the elk genome, and find that elk is likely missing several immune genes.

The authors basically indicate that these missing genes might be the cause of the poor respons to vaccination in elk. It would have been good to see more discussion around this topic when you do raise it. Do you actually pin-point the genetic basis for this inherent difference, or did you just find some missing genes that happened to be immune genes (and no relation to vaccination)?

## Specific comments ##

Line 57: Comma should be after the reference.

Comma is after the reference now. 

Genome assembly:

It is common to polish with PacBio reads using arrow (https://github.com/PacificBiosciences/GenomicConsensus). Why did you not do this? If you had done this on the scaffolds/pseudochromosomes, you might have closed some smaller gaps even.

We polished with Pilon, which allowed the use of both Illumina and Pacbio reads for polishing, Line 137. 

Line 115: How much coverage did you have with CCS reads? It is not clear to me (maybe I missed it) exactly what kind of PacBio library you created. I assume you tried to get the longest reads possible (please add information about this to the manuscript), and then you’d likely not have much coverage in CCS reads.

We did not have great coverage with the CCS reads, 0.72x. I am dumbfounded that I left the tables out of the submission, but it is in supplemental table 2.

Line 125: Or do you mean “the mitochondrion»?

The text now reads mitochondrion. 

Line 129: How do you correct the scaffold with samtools? Just samtools faidx and coordinates?

Sorry, I improved the clarity here. “The scaffold was manually corrected genomic coordinates with faidx in Samtools 1.9 (31).”

Line 148: Do you mean that you ran this through Mikado once more? Interesting approach that results in quite conservative gene annotation I would assume.

Yes, I ran the Mikado annotation pipeline twice. The first round was to get the best gene models possible from the transcriptomic data, just to use as a basis for filtering genes by quality (i.e. expression and repetitiveness). The second round used all of the high confidence genes I identified in the first round, which still provided a high overlap with cattle and red deer gene predictions. There are still ~1000 genes in elk that were not conserved between these species. I have also made the github repo open to the public now, which lays out this decision-making in fine detail. 

Line 182-3: You didn’t map the CLRs to the genome? They would likely map at a lower rate, but could be interesting to see.

We didn’t map the CLRs as the CCS reads are the corrected reads used in the assembly. We wanted to use the percentage of input reads (CCS) that mapped to the final assembly as a measure of assembly quality.

Line 233: Loss of genes is always a bit iffy to discuss. For instance, while you did not find these genes in the genome, were they also lacking from the reads? (Or from the transcriptome reads.) Or, did you look at the (micro)synteny in the affected regions and saw that where you expected the genes to be, there were none and no gaps or otherwise suspiscious sequence? I would like either of these investigations to be done. So, either confirm that the genes are also lacking from the raw reads, or confirm that they are not found based on synteny.

This was a major concern from both reviewers, so we performed a couple of additional analyses that involved confirming the localization of these genes, which we discovered was an error. The naming schemes of these IMGT genes had chromosomal positions within them, and thus were mistaken for their actual mapping position in the cattle genome. Once this was fixed, we were left with 4 genes across 5 regions of the genome that were not near regions of chromosomal fission. Because of this, we changed the title and removed text discussing this. 

We then assessed the RNA-seq expression for these genes/regions in the presence of the elk genome, and found that there was zero expression for these genes/regions. The same analysis was performed with the genes using the Pacbio subreads, and found that three of four genes did not map a pacbio subread, and the one gene that did is nearly 100kb in size. With the absence of expression and the almost complete lack of pacbio overlap, these genes are most likely missing biologically from the elk genome. We have added considerable text to discuss these new analyses. 

Line 246: How would the lack of these genes be utilized in developing a vaccine? I don’t know myself, but when you state this, I would like a bit more elaboration. One approach would be to get transcription data from infected/non-infected individuals to see what the immune system actually does.

Several lines have been added on 283 to 296 explaining the important role of gamma delta T cells in ruminants and how loss of several joining genes could negative impact T cell receptor responses. 

Line 258: The github site for all programs and scripts were not available when I tried accessing it (27th November). It is great that you provide the scripts and such in that way, but unfortunate that I could not go in and browse the repository. The SRA project is also not available (PRJNA657053).

I have made the github repository public for the reviewers’ convenience, I apologize for not having this done sooner. The NCBI data will release automatically as soon as the paper is published online, though I am not sure if it is standard to see the bioproject prior to publication. 

Reviewer #2: The Masonbrink et al. work describes a chromosome-level genome assembly for the Rocky Mountain elk. This new resource will be extremely valuable for the understanding of the elk's immune system and consequently to the prevention of the spread of brucellosis from elk to cattle.

The manuscript is well written and is easy to follow and the putative loss of immune-related genes in the elk show promise for better understand the differences between elk and cattle immune system.

I do have a few comments that I hope will help further improve it.

My main comment relates to the identification of putative gene losses. The result is very interesting and promising. Nonetheless, I believe this section needs a couple of extra checks to support these findings, as these are highlighted in the title and will set this work apart from a regular genome assembly report. For example, could any DNA or RNA seq reads be mapped to the cattle gene sequence? 

Since many of the putative missing genes are at the end of cattle chromosomes they could be within repetitive regions that are more difficult to assemble, but these sequences could still be recovered in the raw data. Other tool the authors could consider using is TOGA (https://github.com/hillerlab/TOGA) which uses pairwise genome alignments to infer orthologous genes between related species and to accurately distinguish orthologs from paralogs or processed pseudogenes. This could be useful to identify genes that were completely lost from pseudogenes, and also identify the mutations which could have led to pseudogenization.

This was a major concern from both reviewers, so we performed a couple of additional analyses that involved confirming the localization of these genes, which we discovered was an error. The naming schemes of these IMGT genes had chromosomal positions within them, and thus were mistaken for their actual mapping position in the cattle genome. Once this was fixed, we were left with 4 genes across 5 regions of the genome that were not near regions of chromosomal fission. Because of this, we changed the title and removed text discussing this. 

We then assessed the RNA-seq expression for these genes/regions in the presence of the elk genome, and found that there was zero expression for these genes/regions. The same analysis was performed with the genes using the Pacbio subreads, and found that three of four genes did not map a pacbio subread, and the one gene that did is nearly 100kb in size. With the absence of expression and the almost complete lack of pacbio overlap, these genes are most likely missing biologically from the elk genome. We have added considerable text to discuss these new analyses. 

Other comments:

1) Page 6 line 132: the parameters used for repeat annotation are not indicated.

I have updated the text to reflect this, “A final version of predicted repeats was obtained using –sensitive 1 and –anno 1 for EDTA 1.7.9 (35) and with default parameters for RepeatModeler 1.0.8 (19) with RepeatMasker 4.1.0(20).”

2) Page 6 line 135: RNA-seq library prep and tissues/cells used are not mentioned in the methods section. The tissues used are listed in the results (page 10 line 201), but I believe they should also be indicated in this section.

We have chosen to add the methods section for the RNA-seq library preparation in the sequencing section (pg 5, lines 102-112) after addition of more information on preparation of DNA for PacBio and Illumina sequencing (lines 91-101). We also added much greater detail to the Animal selection section of the methods to make the data used more transparent. We hope this section addresses the reviewer’s concerns. 

3) Page 8 line 175: is the number of assembled molecules the expected number of chromosomes of the elk? Maybe a citation to a cytogenetics work with that information could be added (e.g. Koulischer, L., et al. (1972). Mammalian cytogenetics. VII. The chromosomes of Cervus canadensis, Elaphurus davidianus, Cervus nippon (Temminck) and Pudu pudu. Acta Zoologica et Pathologica Antverpiensia, 56, 25-30).

An excellent suggestion, something I had forgotten to do. Here is the modified context “Through collaboration with Dovetail Genomics and then additional implementation of the Juicer/JuiceBox/3D-DNA pipeline(16, 17, 22), we generated an assembly of 33 autosomes, an X chromosome, a Y chromosome, a mitochondrial genome, and 151 unincorporated contigs. This result is supported by published cytological studies revealing a haploid set of 34 chromosomes (59).”

59. Koulischer L, Tyskens J, Mortelmans J. Mammalian cytogenetics. VII. The chromosomes of Cervus canadensis, Elaphurus davidianus, Cervus nippon (Temminck) and Pudu pudu. Acta zoologica et pathologica Antverpiensia. 1972;56:25.

4) Page 8 line 185: is there a reason behind not using the mammalian busco set? If used it might give the opportunity to compare the completeness of the assembled genome to that of C. elaphus.

This is because BUSCO4 can be automatically set to the lineage best suited to your species. Eukaryota and cetartiodactyla was automatically selected. 

5) The reviewer pack did not contain supplemental tables so I could not review those.

As I said earlier, I am dumbfounded that I forgot to include these. They are included now. I apologize.

6) There are a few typos or multiple versions of the same word across the manuscript that should be fixed for consistency (e.g. Hi-C vs HI-C vs HiC; elk vs Elk; missing spaces after references; extra commas).

I was able to make corrections to the Hi-C and elk to have consistent naming conventions. I reread and extensively edited the paper again to identify comma and spacing mistakes. These issues are likely fixed.

---

## [Decision Letter · Decision Letter 1]

4 Mar 2021

PONE-D-20-33251R1

A pseudomolecule assembly of the Rocky Mountain elk genome

PLOS ONE

Dear Dr. Masonbrink,

Thank you for submitting your manuscript to PLOS ONE. After careful consideration, we feel that it has merit but does not fully meet PLOS ONE’s publication criteria as it currently stands. Therefore, we invite you to submit a revised version of the manuscript that addresses the points raised during the review process.

Please address Reviewer' #2's remaining comment.  Almost there!

We look forward to receiving your revised manuscript.

Kind regards,

F. Alex Feltus, Ph.D.

Academic Editor

PLOS ONE

Journal Requirements:

Reviewers' comments:

Reviewer's Responses to Questions

**Comments to the Author**

1. If the authors have adequately addressed your comments raised in a previous round of review and you feel that this manuscript is now acceptable for publication, you may indicate that here to bypass the “Comments to the Author” section, enter your conflict of interest statement in the “Confidential to Editor” section, and submit your "Accept" recommendation.

Reviewer #1: All comments have been addressed

Reviewer #2: (No Response)

2. Is the manuscript technically sound, and do the data support the conclusions?

Reviewer #1: Yes

Reviewer #2: Yes

3. Has the statistical analysis been performed appropriately and rigorously? 

Reviewer #1: Yes

Reviewer #2: N/A

4. Have the authors made all data underlying the findings in their manuscript fully available?

Reviewer #1: No

Reviewer #2: Yes

5. Is the manuscript presented in an intelligible fashion and written in standard English?

Reviewer #1: Yes

Reviewer #2: Yes

6. Review Comments to the Author

Reviewer #1: (No Response)

Reviewer #2: My concerns have mostly been addressed.

I believe section describing the putative gene loss could still be further improved. For example, how does the elk genome sequence look on the regions these genes were expected to locate? Are there gaps? Are these contiguous regions? It would also be very interesting to see if there are still remnants of these genes in the elk genome? Or are these genes located at the boundaries of chromosomal inversions, for example? Nonetheless, I do understand that the last two questions might require more time to investigate. The information about the sequence contiguity around these putative lost genes, however, would address the concerns of misassembly in these regions.

7. PLOS authors have the option to publish the peer review history of their article (what does this mean?). If published, this will include your full peer review and any attached files.

Reviewer #1: No

Reviewer #2: **Yes: **Joana Damas

---

## [Author Response · Author response to Decision Letter 1]

25 Mar 2021

Dear Editor,

We thank the reviewers for their time and effort in reviewing this manuscript, as well as improving the clarity of our analyses. Please see directed comments below.

Sincerely,

Rick Masonbrink

I believe section describing the putative gene loss could still be further improved. For example, how does the elk genome sequence look on the regions these genes were expected to locate? Are there gaps? Are these contiguous regions? It would also be very interesting to see if there are still remnants of these genes in the elk genome? Or are these genes located at the boundaries of chromosomal inversions, for example? Nonetheless, I do understand that the last two questions might require more time to investigate. The information about the sequence contiguity around these putative lost genes, however, would address the concerns of misassembly in these regions.

We have refurbished the gene loss analysis to have more clarity and direct interpretations from the data. The new analysis includes the sequence information of these gene loss regions, and the corresponding region for cattle (Table s6). Initially we investigated these gene losses with just tBLASTn, but have since investigated these regions extensively with BLASTn, adding significant depth to the information on these regions. Please see the methods at line 189-196 to see that tBLASTn, blastn of cattle sequences that hit but were missing in elk, and blastn of cattle sequence hits + 20bp on each side BLastn to elk, and then extracted those sequences in cattle with +100bp borders surrounding for mapping RNAseq and Pacbio reads. These analyses definitely added depth to where these genes may have gone, how they were modified, and/or lost. However, we did not find the remnants or borders of these genes at the borders of chromosomal fissions/fusions or with a verified chromosomal rearrangements. We hope that this additional analysis will be sufficient to allay reviewer concerns about misassembly and/or an incomplete analysis.

---

## [Editor Report · Decision Letter 2]

29 Mar 2021

A pseudomolecule assembly of the Rocky Mountain elk genome

PONE-D-20-33251R2

Dear Dr. Masonbrink,

We’re pleased to inform you that your manuscript has been judged scientifically suitable for publication and will be formally accepted for publication once it meets all outstanding technical requirements.

Kind regards,

F. Alex Feltus, Ph.D.

Academic Editor

PLOS ONE
---

## [Editor Report · Acceptance letter]

13 Apr 2021

PONE-D-20-33251R2 

A pseudomolecule assembly of the Rocky Mountain elk genome  

Dear Dr. Masonbrink:

I'm pleased to inform you that your manuscript has been deemed suitable for publication in PLOS ONE. Congratulations! Your manuscript is now with our production department. 

Kind regards, 

on behalf of

Dr. F. Alex Feltus 

Academic Editor

PLOS ONE